# Flexible Wet-Spun PEDOT:PSS Microfibers Integrating Thermal-Sensing and Joule Heating Functions for Smart Textiles

**DOI:** 10.3390/polym15163432

**Published:** 2023-08-17

**Authors:** Yan Li, Hongwei Hu, Teddy Salim, Guanggui Cheng, Yeng Ming Lam, Jianning Ding

**Affiliations:** 1School of Mechanical Engineering, Jiangsu University of Science and Technology, Zhenjiang 212003, China; yanli@just.edu.cn; 2Institute of Intelligent Flexible Mechatronics, School of Mechanical Engineering, Jiangsu University, Zhenjiang 212013, China; dingjn@ujs.edu.cn; 3School of Materials Science and Engineering, Nanyang Technological University, 50 Nanyang Avenue, Singapore 639798, Singapore; tsalim@ntu.edu.sg (T.S.); ymlam@ntu.edu.sg (Y.M.L.); 4School of Mechanical Engineering, Yangzhou University, Yangzhou 225009, China

**Keywords:** thermoelectric fiber, conductive polymer, smart textile, temperature sensor, PEDOT:PSS

## Abstract

Multifunctional fiber materials play a key role in the field of smart textiles. Temperature sensing and active thermal management are two important functions of smart fabrics, but few studies have combined both functions in a single fiber material. In this work, we demonstrate a temperature-sensing and in situ heating functionalized conductive polymer microfiber by exploiting its high electrical conductivity and thermoelectric properties. The conductive polymer microfibers were prepared by wet-spinning the PEDOT:PSS aqueous dispersion with ionic liquid additives, which was used to enhance the electrical and mechanical properties of the final microfibers. The thermoelectric properties of these microfibers were further studied. Due to their excellent flexibility and mechanical properties, these fibers can be easily integrated into commercial fabrics for the manufacture of smart textiles through knitting. We further demonstrated a smart glove with integrated temperature-sensing and in situ heating functions, and further explored thermoelectric fiber-based temperature-sensing array fabric. These works combine the thermoelectric properties and heating function of conductive polymer fibers, providing new insights that enable further development of high-performance, multifunctional wearable smart textiles.

## 1. Introduction

Smart textiles can not only sense environmental changes, but also automatically respond to the surrounding environment or stimuli, such as thermal, chemical, or mechanical changes [1]. They have a wide application prospect in sensing human physiological signals at close range, supporting and assisting daily human activities [2,3,4]. The development of multifunctional fiber materials is the key to ensuring the functionality and practicality of smart fabrics. Currently, electronic fibers can already be endowed with functions such as high electrical conductivity, strain response, pressure sensing, luminescence, photoelectric sensing, etc. [5,6,7,8]. Most of those studies focus on optimizing only a specific function, while research on intricate multifunctional integrated fibers, which could greatly reduce the complexity of device fabrication and system construction, remains scarce.

Conductive polymers (CPs) have received much attention due to their excellent flexibility, electrical properties, and electrochemical activity [9,10]. Poly(3,4-ethylenedioxythiophene)/polystyrene sulfonate (PEDOT/PSS) is one of the most widely used CPs as an intrinsically flexible electronic material, showing great promise in elastic conductors, sensing, displays, and other applications [11,12]. PEDOT:PSS can be attached to fabrics by printing, dyeing, and in situ polymerization to fabricate multifunctional electronic fabrics [13]. In recent years, researchers have prepared pure PEDOT:PSS fibers by wet-spinning, and obtained PEDOT:PSS fibers with high electrical conductivity and excellent mechanical properties by adjusting the spinning dope, coagulation bath, and post-treatment [14,15,16,17,18]. These fibers are further used to prepare highly sensitive sensors, supercapacitors, and organic electrochemical transistors [14,17,18]. On the other hand, these fibers have thermoelectric properties and can be used to harvest wearable energy directly from the heat emitted by the body [19,20,21,22]. These studies inspired us to develop a multifunctional fiber based on PEDOT:PSS. Herein, we explored the temperature-sensing and Joule heating functionalized PEDOT:PSS fiber fabric by exploiting their high electrical conductivity and thermoelectric properties. Firstly, PEDOT:PSS fibers were prepared by wet-spinning PEDOT:PSS dispersion with ionic liquid additives, which was used to improve the electrical conductivity and mechanical properties. After studying the thermoelectric properties of these fibers, we further demonstrated a temperature-sensing fabric array based on PEDOT:PSS fibers and its in situ heating capabilities. A temperature sensor array composed of a few simple interlaced fibers was further fabricated to demonstrate their versatile applications. Finally, we demonstrate that thin polymer sheaths can greatly improve the working stability of PEDOT:PSS fibers in different environments. This work integrates the thermoelectric properties and heating functions of conductive polymer fibers, providing a new idea for further development of high-performance, multifunctional wearable smart fabrics.

## 2. Materials and Methods

### 2.1. Materials

Ionic liquid 1-butyl-3-methylimidazolium tosylate (BMImOTs, 99%) was purchased from Lanzhou Greenchem ILs (Lanzhou, China); PEDOT:PSS aqueous solution (solid content 1.0–1.3 wt%, PEDOT to PSS ratio 1:2.5, conductivity 700~800 S/cm) was purchased from Shanghai Ouyi Organic Optoelectronic Materials (Shanghai, China).

### 2.2. Wet-Spinning of PEDOT:PSS Fibers

The PEDOT:PSS (PP) microfibers were fabricated through a wet-spinning process as illustrated in Figure 1a. To enhance the electrical conductivity and mechanical properties, a specifically selected ionic liquid (BMImOTs) was first added to the PP spinning dope to reduce the electrostatic interaction between PEDOT and PSS. The spinning dope was prepared by adding BMImOTs (0–2.6 μg) into the PEDOT:PSS solution (10 mL), so that the BMImOTs accounted for 0 to 5 wt% of the solid content. Then, the solution was rotated and evaporated at 50 °C to obtain PEDOT:PSS solution with concentration of 2.0 to 2.6 wt%. All dispersions were bath-sonicated for 30 min to remove the bubbles prior to fiber spinning. The spinning dope was then loaded into a 3 mL syringe and injected into the sulfuric acid coagulation bath through a needle with an inner diameter of 400 μm. In the sulfuric acid coagulation bath, water and a portion of the PSS in the solution were removed, forming PP microfiber. After entering the coagulation bath, the fibers were immediately collected on the relay roller to prevent fiber entanglement. The fibers were then soaked in the coagulation bath for 3 h before being pulled through a washing bath containing ethanol/water (volume ratio of 3:1) to remove residual sulfuric acid. The fibers were then drawn through two heating plates and finally collected on the roller to obtain PEDOT:PSS fibers. The wet-spinning process used is applicable for continuous production, and in principle, the length of the filament is only limited by the amount of spinning solution. Figure 1b shows ~50 m long microfibers, which have excellent flexibility and can be stitched onto fabrics for use in wearable smart fabrics. Figure 1c shows a microfiber lifting a weight of 10 g, indicating that the microfiber has good mechanical properties.

### 2.3. Characterization

The mechanical properties of the fibers were measured using a thermomechanical analysis system (TMA, Hitachi 7100E, Hitachi, Tokyo, Japan). A single fiber was used for the tensile strain-stress measurement. The length of the tested fiber sample was approximately 2 cm, and the cross-section area was determined by scanning electron microscope (SEM) image. The electrical conductivity of the PEDOT:PSS fiber was measured using a source meter (Keithley, 2450 SourceMeter, Tektronics, Beaverton, OR, USA). Samples were prepared by placing the fiber on a glass slide with the two ends of the fiber connected to copper wires using silver paste. A low voltage (1–3 V) was applied to the two ends and the resistance was recorded. The conductivity was then calculated by taking into account the sample length and cross area. The Seebeck coefficient was characterized using a home-made device as shown in the Appendix A. Two Peltier modules were used for regulating the temperature difference of the two ends of the fiber, while the thermovoltage was recorded using Keithley 2450. For the above mechanical, electrical, and thermoelectric measurements, at least 10 samples from the same batch of fibers were tested and the average values were obtained. The morphology of the fibers was studied using SEM images (JSM-IT800, JEOL, Tokyo, Japan). Multiple fibers were spread on conductive carbon tape without further treatment. The fibers were cut using scissors for taking the cross-sectional SEM images. The X-ray diffraction (XRD) and wide-angle X-ray scattering (WAXS) were acquired on D8 Advance (Bruker AXS GmbH, Karlsruhe, Germany) and Nano-inXider (Xenocs SAS, Grenoble, France), respectively.

### 2.4. Preparation of Smart Textiles and Stability Testing

Smart fabrics with thermal-sensing and Joule heating functions were manufactured by weaving PEDOT:PSS microfiber onto commercial gloves. Firstly, ten microfibers were twisted to form a stable fiber yarn. Then, multiple-fiber yarns were sewn onto the glove with a sewing needle. The sensing unit on the index finger consisted of three strands of yarn that detect temperature changes when grasping objects. The two ends of the yarn were placed on the inside and outside of the glove, fixed with silver paste and copper wire, and connected to the thermoelectric potential measurement system. The sensing unit on the back of the glove consisted of six strands of yarn to detect the ambient temperature and provide heating. One end of the bundle was positioned close to the skin, while the other end was exposed to the environment to sense the temperature difference between the environment and the skin of the hand.

To make the temperature sensor array, six bundles of yarn were sewn onto the fabric in parallel three lines, while using silver paste and copper wire to connect both ends of each bundle to a multichannel voltage signal acquisition board.

In order to improve the working stability of the fibers, a thin layer of PDMS was wrapped on the surface of a yarn bundle consisting of 20 PP fibers. The twisted yarn bundle was slowly pulled through the freshly formulated PDMS precursor (Sylgard 184, Dow Corning, Midland, MI, USA). After standing vertically for 15 min, the fiber bundles were treated in an oven at 80 °C for half an hour. The moisture stability of the yarn was tested by dripping water in the middle of the yarn while the conductivity and Seebeck coefficient were measured. The PDMS-wrapped yarn was placed in water and stirred for one hour at a rate of 500 rpm for one wash cycle. After each wash, the conductivity of the fibers and the Seebeck coefficient were measured.

## 3. Results and Discussion

The morphology and structural characteristics of microfibers have significant influence on their mechanical, electrical, and thermoelectric properties. Figure 2a shows the SEM image of the microfibers, from which it can be seen that the microfibers are in fact mainly flat ribbons with a width of about 40 to 60 μm. Figure 2b shows the cross-section of a single fiber, with a pronounced feature of a flattened ribbon. The microfiber is about 60 μm wide and 10 μm thick, which translates into a cross-sectional area of about 600 μm^2^. The cross-sectional size of the fiber is mainly determined by the dope concentration and the diameter of the injection needle. After rounds of optimization, the needle with a diameter of 400 μm was found to provide fibers with the best mechanical and electrical properties, and was subsequently used throughout the study. The rapid water loss after the injection into the coagulation bath prompts a structural collapse, leading to the formation of flat fibers. Compared with the hollow structure formed by PP fibers in previous studies [23], the presence of ionic liquid in the precursor solution in our work could facilitate a stronger interpolymer interaction and generate a network structure, thus forming dense flat fibers, which is beneficial to improving the mechanical properties of fibers.

The XRD pattern of the fiber (Figure 2c) clearly shows the crystalline regions as evident in the (100) and (200) diffraction peaks originated from the alternate stacking of PEDOT and PSS, and π-π stacking of the crystallized PEDOT chains (010), which proves that PEDOT and PSS in the fiber have good crystallinity [24,25]. In the WAXS image shown in Figure 2d, the corresponding scattering patterns can also be seen, the plot of which is shown in the Appendix A. Furthermore, these diffraction peaks are mainly located along the radial direction of the fiber, indicating that the fiber has a high degree of anisotropy, which is beneficial to improving the mechanical properties of the fiber in the axial direction [14].

The mechanical properties of the fibers were further tested, and their tensile stress curves are shown in Figure 3a. The maximum strain and strength of the fibers are shown in Figure 3b. It can be seen that the addition of ionic liquid has a significant effect on the mechanical properties of the prepared fibers. A maximum strain of 19% with a breaking strength approaching 206 MPa could be obtained with 2% ionic liquid addition. The specimens nearly doubled their maximum strain and breaking strength compared to those without additives. This result indicates that the addition of ionic liquid can improve the microstructure of the fibers and hence its mechanical properties, which is similar to the effect on PP thin films investigated by Wang et al. [26]. The addition of ionic liquid can reduce the electrostatic interaction between PEDOT and PSS groups, thereby increasing the cross-linking degree of PEDOT and further forming a film with high mechanical properties. However, adding too much ionic liquid could have a detrimental effect, as can be seen from the premature failure of the PP fiber, much sooner than the pristine fiber.

The addition of ionic liquids also significantly improved the electrical properties of the fibers. Figure 4a shows the electrical conductivity of fibers formed from different ionic liquids. It can be seen that by adding 2% ionic liquid, the conductivity of the fiber is significantly increased from 1170 S/cm to 2258 S/cm. The trends in conductivity are generally similar to those in mechanical properties. It is plausible that the structural changes responsible for the improvement in mechanical properties also have direct consequences on the electrical properties. We further tested the thermoelectric properties of the fibers. Figure 4b shows the Seebeck coefficient and power coefficient of the fibers. The Seebeck coefficient of fibers with ionic liquid addition showed a downward trend with the increase in ionic liquid addition, mainly because the doping level of PEDOT was increased with the addition of ionic liquid, thereby reducing its Seebeck coefficient [27]. The power factor of the fiber, calculated from the conductivity and the Seebeck coefficient, is shown in Figure 4b, from which it can be seen that the power factor is mainly dominated by the conductivity of the fiber, that is, at the optimum ionic liquid concentration of 2%, the power coefficient also reaches a maximum value of 90 μW/m·K^2^. Therefore, due to their excellent mechanical and electrical properties, we selected fibers with 2% ionic liquids to be integrated into the prototype device in the follow-up studies. Figure 4c shows the thermoelectric potential of the fibers at different temperature differences. It can be seen that the thermoelectric potential exhibits a strong linear relationship with the temperature difference, which is highly favorable for temperature-sensing applications which will be discussed later. We further tested the thermoelectric output properties of the fibers, as shown in Figure 4d. At a temperature difference of 20 °C, the open-circuit voltage and short-circuit current of the fiber reached 0.42 mV and 32 μA, respectively, and its maximum power reached 3.5 nW.

To demonstrate the application of PP fibers in wearable smart sensing, we weave the PP microfiber bundles into gloves, as shown in Figure 5a. Due to the good flexibility, the microfibers fit well with the existing glove fibers. Three PP microfiber bundles are woven on the inside of the glove fingers, and six bundles on the back of the hand. The two ends of each fiber and the contact electrodes are, respectively, distributed on the inner and outer sides of the glove to realize the temperature measurement function by using the temperature difference between both ends. The fibers on the finger can measure the temperature difference between the object and the hand when grasping the object, while the fibers on the back of the hand can not only measure the temperature difference between the environment and the hand itself, but it can also provide a heating function when needed. Figure 5b shows the responses of the glove to the cold and hot water cups. It can be seen that the thermoelectric potential generated by the fiber sensor has a good corresponding relationship with the temperature, and the response speed is almost synchronous. Figure 5c shows the thermoelectric potential of the glove at an ambient temperature of 18 °C. Since the temperature of the human hand is relatively constant, the ambient temperature can be easily inferred from the reading of the thermoelectric potential. On the other hand, due to the relatively low resistance and good environmental stability of PP fibers, PP fibers can also achieve Joule heating by applying a bias voltage. Figure 5d shows the temperature change that occurs when a voltage of 1 V to 5 V was applied to the PP fibers on a smart glove, where the ambient temperature was about 0 °C. The higher the voltage, the more pronounced the Joule heating effect. When voltages of 3 V, 4 V, and 5 V were applied, the surface temperature could be increased to 11 °C, 21 °C, and 38 °C, respectively. This suggests that these low-voltage electrical heating fabrics can be combined with commercial portable battery packs to provide heat to the human body in low-temperature environments. Combining the thermoelectric sensing function of the fiber with the heating function, our PP-fiber-embedded prototype gloves can activate the heating function upon sensing a low ambient temperature, thereby providing a timely thermal protection for the human hand.

Fiber-shaped thermoelectric materials offer a new possibility for building smart fabrics with temperature-location sensing. Both ends of the fiber can simultaneously sense the temperature changes, and then sense where the temperature changes occur. Taking advantage of this feature of PP fibers, we constructed a 3 × 3 sensing array for position and temperature sensing, as shown in Figure 6a. The position points are connected by fibers, and by analyzing and comparing the voltage signals collected by the fibers, the positions of the sensed temperature and the temperature difference value can be obtained. Figure 6b shows the thermoelectric potential of the fibers when the finger touches nine different locations (labeled A to I). It can be seen that touching points A and D will generate opposite thermoelectric potentials at the two ends of fiber 1; while touching point D, a thermoelectric potential will also be generated on fiber 4. Therefore, the location and temperature of the object can be restored by comparing the thermoelectric potential generated at different points. Figure 6c shows a hot metal rod and two cold metal weights placed on the thermoelectric fiber fabric, and Figure 6d shows the temperature at nine points on the fabric deduced from the generated thermoelectric potential, which is consistent with the measured results. This result provides a new idea for the further construction of intelligent sensing arrays of thermoelectric fibers.

The electrical properties of conductive polymers are often influenced by ambient humidity, which affects their working stability. In this work, since the thermoelectric and heating functions of conductive polymer fibers can be carried out by means of heat exchange, they do not need to be exposed directly to the environment. We wrap a thin layer of hydrophobic PDMS outside the PP fiber yarn to reduce the influence of moisture on the fiber properties. As shown in Figure 7a, PDMS forms an extremely thin protective layer (about a few microns) around the fiber yarn, which does not seriously affect the heat conduction between the PP fiber and the external environment. As shown in Figure 7b and c, the conductivity and Seebeck coefficient of uncoated PP fibers decreased significantly when wet. In contrast, the conductivity and thermoelectric coefficient of the wrapped PP fiber, even if immersed in water, are hardly affected. We then washed the wrapped fibers several times in water. As shown in Figure 7d, after five cycles of washing, the conductivity and thermoelectric coefficient of the fiber remain above 95% of the original value. The margin loss may be due to the penetration of trace amounts of water molecules into the fiber from the crack or junction. Nevertheless, we believe that PDMS wrapping can significantly improve the working stability of PP fibers, providing a material basis for the realization of multifunctional smart fabrics.

## 4. Conclusions

In summary, a temperature-sensing and in situ heating functionalized microfiber fabric has been demonstrated by exploiting the high electrical conductivity and thermoelectric properties of the conductive polymer microfibers. By adding ionic liquid into the wet-spinning dope, both the conductivity and mechanical properties of the fabricated microfibers were improved. We further demonstrated a multifunctional fabric based on conductive polymer microfibers equipped with both temperature-sensing and in situ heating capabilities. In addition, a sensor array consisting of a few simple interlaced microfibers was manufactured to collect information on the temperature and position of the object. This work integrates the thermoelectric properties and heating functions of PEDOT:PSS, providing a new idea for further development of high-performance, multifunctional wearable smart fabrics.

## Figures and Tables

**Figure 1 polymers-15-03432-f001:**
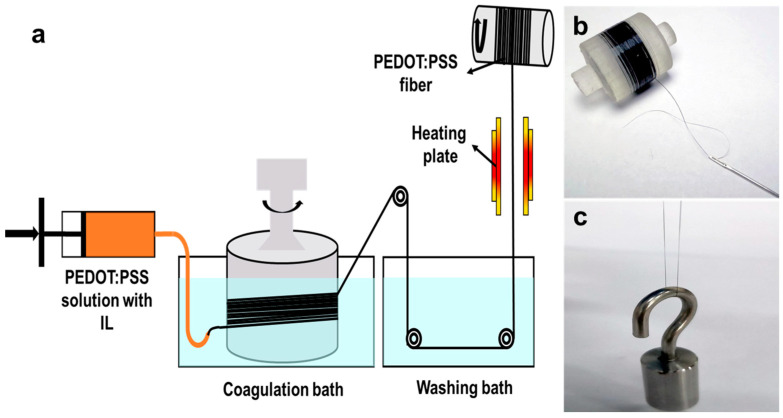
Fabrication process of the PP microfiber using a wet-spinning method (**a**), photo of a roll of PP microfiber (**b**), and a microfiber lifting 10 g weight (**c**).

**Figure 2 polymers-15-03432-f002:**
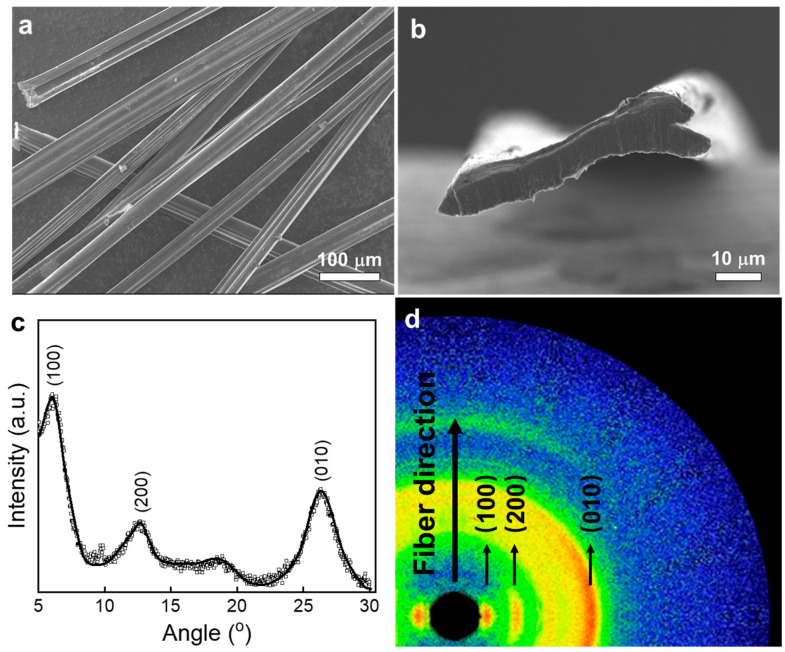
Characterization of the PP fibers: SEM images of the multiple fibers (**a**) and a cross-section of the fiber (**b**), XRD pattern of the fibers (**c**), and WAXS image of the fiber (**d**).

**Figure 3 polymers-15-03432-f003:**
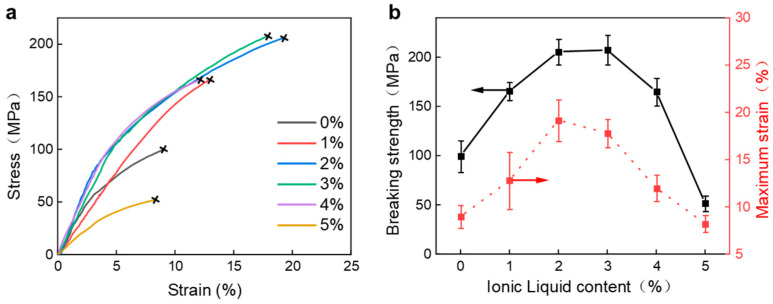
Mechanical properties of the PP fiber: stress–strain curves of the PP fibers with varied ionic liquid content (**a**), breaking strength and maximum strain for the PP fibers (**b**).

**Figure 4 polymers-15-03432-f004:**
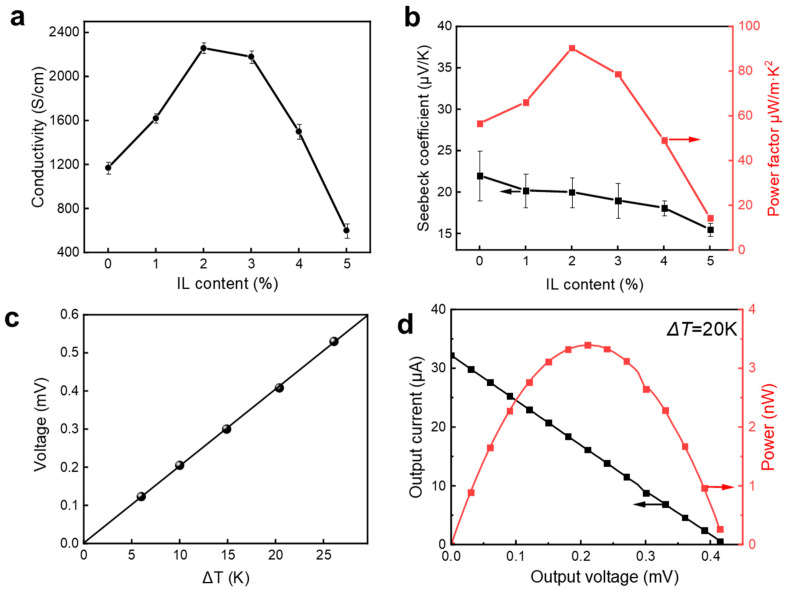
Electrical and thermoelectric properties of PP fibers: electrical conductivity (**a**), Seebeck co-efficient and power factor (**b**), thermoelectric voltage at varied DT (**c**), and output power at Δ*T* = 20 K (**d**).

**Figure 5 polymers-15-03432-f005:**
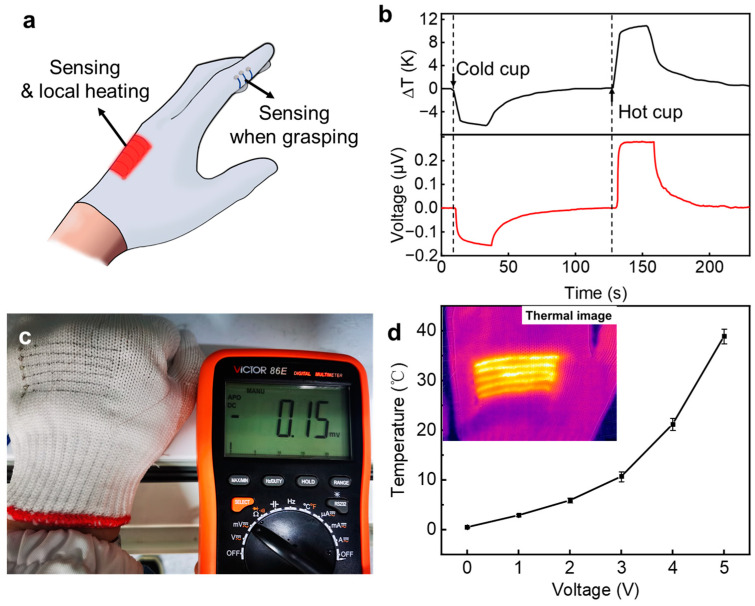
Demonstration of smart glove based on PP fibers: multiple sensing and heating functions integrated on fabric glove (**a**), sensing of cold cup and hot cup when grasping (**b**), thermoelectric voltage generated when temperature drops to 18 °C (**c**), temperature raising by applying various voltages to the PP fibers (**d**), inset shows the thermal image of smart glove with heating area.

**Figure 6 polymers-15-03432-f006:**
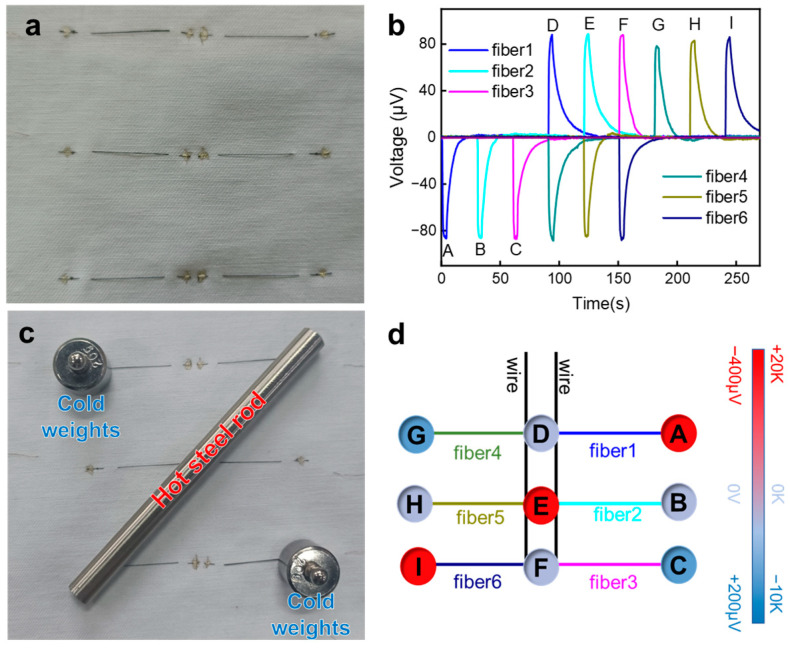
Thermoelectric fiber array for location and temperature sensing: distribution of six fibers and the nine sensing points (**a**), thermoelectric voltages generated from the nine points (labeled from A to I) when touched with finger (**b**), sensing of multiple objects with varied temperatures (**c**), and their location and temperature obtained from the sensing array (**d**).

**Figure 7 polymers-15-03432-f007:**
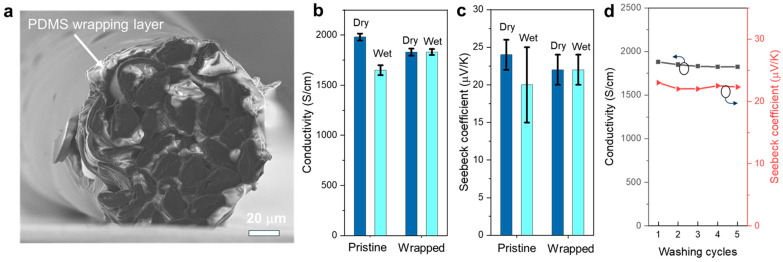
Improving the stability of PP fibers by PDMS wrapping: cross-sectional SEM image showing the PDMS wrapping on the PP fiber yarns (**a**), conductivity (**b**) and Seebeck coefficient (**c**) of the pristine PP fibers and wrapped PP fibers in dry and wet conditions, and the performance change of the PP fibers after multiple washing cycles (**d**).

## Data Availability

The data that support the findings of this study are available from the authors upon reasonable request.

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
