# Peer review of "Flexible Wet-Spun PEDOT:PSS Microfibers Integrating Thermal-Sensing and Joule Heating Functions for Smart Textiles"

_polymers, 2023, doi:10.3390/polym15163432_

Round 1

Reviewer 1 Report

The present article reports fabrication of PEDOT:PSS fibers containing ionic liquid by wet-spinning, followed by its weaving onto commercial gloves in order to obtain smart textile for temperature sensing and local heating.

In order to improve the quality of the article I recommend minor revision before publication.

My remarks are detailed below:

1. I recommend modification of the TITLE in order to highlight the emphasis of the study with more detailed information.

2. In Abstract I recommend to include information on the composition of fibers and preparation of smart textile.

3. The introduction could benefit from more emphasis on the significance and novelty of the research. To enhance this, it may be helpful to provide more context and background information that explains why the research is important and how it contributes to the existing literature. Please, compare your work with well-known in Refs 12 and 24.

4. In characterization section – detailed information for tensile specimens (size and form) and conditions for mechanical tests, as well for electrical conductivity measurements and SEM analysis (specimen preparation) are missed.

5. In my opinion section Discussion could be Conclusions.

Author Response

The present article reports fabrication of PEDOT:PSS fibers containing ionic liquid by wet-spinning, followed by its weaving onto commercial gloves in order to obtain smart textile for temperature sensing and local heating.

In order to improve the quality of the article I recommend minor revision before publication.

My remarks are detailed below:

  1. I recommend modification of the TITLE in order to highlight the emphasis of the study with more detailed information.

Response: Thank you for your suggestion. We have revised the title to highlight the emphasis of this study by including more detailed information:

“Flexible Wet-spun PEDOT:PSS Microfibers Integrating Thermal-sensing and Joule Heating Functions for Smart Textiles”

  1. In Abstract I recommend to include information on the composition of fibers and preparation of smart textile.

Response: Thank you for your suggestion. We have modified the abstract as:

“The conductive polymer microfibers were prepared by wet-spinning the PEDOT:PSS aqueous dispersion with ionic liquid additives, which was used to enhance the electrical and mechanical properties of the final microfibers. The thermoelectric properties of these microfibers were further studied. Due to their excellent flexibility and mechanical properties, these fibers can be easily integrated into commercial fabrics by knitting for the manufacture of smart textiles.”

  1. The introduction could benefit from more emphasis on the significance and novelty of the research. To enhance this, it may be helpful to provide more context and background information that explains why the research is important and how it contributes to the existing literature. Please, compare your work with well-known in Refs 12 and 24.

Response: Thank you for your comment. We have rewritten the second paragraph in introduction shown as follows. The background of PEDOT:PSS application in smart textile and the preparation of PEDOT:PSS fibers have been included. This work have two contributions to the previous reports: the use of ionic liquid additive to enhance the properties of final fibers and the integrating of thermoelectric sensor and Joule heating functions into these fibers.

“Conductive polymers (CPs) have received much attention due to their excellent flexibility, electrical properties, and electrochemical activity [9,10]. Poly(3,4-ethylenedioxythiophene)/polystyrene sulfonate (PEDOT/PSS) is one of the most widely used CPs, as an intrinsically flexible electronic material, showing great promise in elastic conductors, sensing, displays, and other applications [11, 12]. PEDOT: PSS can be attached to fabrics by printing, dyeing, and in situ polymerization to fabricate multifunctional electronic fabrics [13]. In recent years, researchers have prepared pure PEDOT:PSS fibers by wet spinning, and obtained PEDOT:PSS fibers with high electrical conductivity and excellent mechanical properties by adjusting the spinning dope, coagulation bath and post-treatment [14-18]. These fibers are further used to prepare highly sensitive sensors, supercapacitors, and organic electrochemical transistors [14, 17, 18]. On the other hand, these fibers have thermoelectric properties and can be used to harvest wearable energy directly from the heat emitted by the body [19, 20]. These studies inspired us to develop a multifunctional fiber based on PEDOT:PSS. Herein, we explored the temperature-sensing and Joule heating functionalized PE-DOT:PSS fiber fabric by exploiting their high electrical conductivity and thermoelectric properties. Firstly, PEDOT:PSS fibers were prepared by wet-spinning PEDOT:PSS dispersion with ionic liquid additives, which was used to improve the electrical conductivity and mechanical properties….”

  1. In characterization section – detailed information for tensile specimens (size and form) and conditions for mechanical tests, as well for electrical conductivity measurements and SEM analysis (specimen preparation) are missed.

Response: Thanks for your comment. We have revised the characterization part as suggested to include more detailed information:

“The mechanical properties of the fibers were measured using a thermomechanical analysis system (TMA, Hitachi 7100E). A single fiber was used for the tensile strain-stress measurement. The length of the tested fiber sample was approximately 2 cm and the cross-section area was determined by scanning electron microscope (SEM) image. The electrical conductivity of the PEDOT:PSS fiber was measured using a source meter (Keithley, 2450 SourcMeter). Samples were prepared by placing the fiber on a glass slide with the two ends of the fiber connected to copper wires using silver paste. A low voltage (1-3 V) was applied to the two ends and the resistance was recorded. The conductivity was then calculated by taking account the sample length and cross area. The Seebeck coefficient was charaterized using a home-made device as shown in the supplementary information. Two peltier modules were used for regulating the tem-perature difference of the two ends of the fiber, while the thermovoltage was recorded using Keithley 2450. For the above mechanical, electrical and thermoelectric meas-urements, at least 10 samples from the same batch fibers were tested and the average values were obtained. The morphology of the fibers was studied using SEM images (JSM-IT800). Multiple fibers were spread on conductive carbon tape without further treatment. The fibers were cut using scissors for taking the cross-sectional SEM images.”   

  1. In my opinion section Discussion could be Conclusions.

Response: Thanks for your comment. We have changed the last section as Conclusions.

Reviewer 2 Report

I read the Authors' work with interest. The publication can be qualified for publication after taking into account significant corrections regarding the scope of the performed investigations. These corrections are needed to allow the Readers to assess whether we are dealing with a real proof of concept material or just a scientific curiosity that will never find application beyond the walls of a laboratory. 1. "generated by the fibers on the glove when a voltage of 1V to 4V is applied, and the temperature can be raised from room temperature 24°C to over 65°C". - At room temperature, the human body feels thermal comfort and it is not necessary to warm the hands. It would be reasonable to make measurements in temperatures below zero degrees Celsius. 2. Conductive polymers show a large change in response as a function of humidity. Please show how humidity affects the operation of the proposed system. This is especially important in the case of smart fabrics, which are expected to come into contact with sweat, where, apart from moisture, we have to deal with salinity 3. How well is the ionic liquid additive bound in smart weaving and whether it does not escape from the fabric? 4. How the performance of the material is affected by multiple washing cycles? 5. Please rearrange the scheme of making the glove in the experimental part. 6. The first paragraph of results and discussion section describes the procedure for producing fibers and will be included in the experimental part of the work. 4. One paragraph of discussion of the results is really not much for a scientific paper.

English is ok.

Author Response

I read the Authors' work with interest. The publication can be qualified for publication after taking into account significant corrections regarding the scope of the performed investigations. These corrections are needed to allow the Readers to assess whether we are dealing with a real proof of concept material or just a scientific curiosity that will never find application beyond the walls of a laboratory.

  1. "generated by the fibers on the glove when a voltage of 1V to 4V is applied, and the temperature can be raised from room temperature 24°C to over 65°C". - At room temperature, the human body feels thermal comfort and it is not necessary to warm the hands. It would be reasonable to make measurements in temperatures below zero degrees Celsius.

Response: Thank you for your comment. Based on your suggestion, we retested the Joule heating effect of these gloves at low temperatures and the results were provided in Figure 5d and the relative discussion were shown in the main text:

“Figure 5d shows the temperature change that occurs when a voltage of 1 V to 5 V was applied to the PP fibers on a smart glove, where the ambient temperature is about 0°C. The higher the voltage, the more pronounced the Joule heating effect. When voltages of 3 V, 4 V and 5V were applied, the surface temperature could be increased to 11°C, 21°C, and 38°C, respectively. This suggests that these low-voltage electrical heating fabrics can be combined with commercial portable battery packs to provide heat to the human body in low temperature environments.”

  1. Conductive polymers show a large change in response as a function of humidity. Please show how humidity affects the operation of the proposed system. This is especially important in the case of smart fabrics, which are expected to come into contact with sweat, where, apart from moisture, we have to deal with salinity.

Response: Thank you for your comment. Indeed, multiple factors, such as humidity and salinity, affect the conductivity and other performance of conductive polymers as suggested by the reviewer and many previous reports. To achieve the stable operation of our proposed functional fabrics, we performed protective wrapping on the PEDOT:PSS fibers to prevent the penetration of moisture. This is similar to any electronic materials and devices that appropriate sealing and protective layers are needed to reach long-term operation stability. We then proved that an ultra-thin polymer (PDMS) wrapping layer on the sheath of PEDOT:PSS fiber yarns can dramatically improve the stability of whole system (conductivity and thermoelectric properties). The relative results were provided in Figure 7 and the corresponding discussion was provided in the main text:

“The electrical properties of conductive polymers are often influenced by ambient humidity, which affects their working stability. In this work, since the thermoelectric and heating functions of conductive polymer fibers can be carried out by means of heat exchange, they do not need to be exposed directly to the environment. We wrap a thin layer of hydrophobic PDMS outside the PP fiber yarn to reduce the influence of moisture on the fiber properties. As shown in Figure 7a, PDMS forms an extremely thin protective layer (about a few microns) around the fiber yarn, which does not seriously affect the heat conduction between the PP fiber and the outside world. As shown in Figure 7 b and c, the conductivity and Seebeck coefficient of uncoated PP fibers de-creased significantly when wet. In contrast, the conductivity and thermoelectric coef-ficient of the wrapped PP fiber, even if immersed in water, are hardly affected. We then washed the wrapped fibers several times in water. As shown in Figure 7d, after 5 cycles of washing, the conductivity and thermoelectric coefficient of the fiber remain above 95% of the original value. The margin loss may be due to the penetration of trace amounts of water molecules into the fiber from the crack or junction. Nevertheless, we believe that PDMS wrapping can significantly improve the working stability of PP fibers, providing a material basis for the realization of multifunctional smart fabrics.”

“Figure 7. Improving the stability of PP fibers by PDMS wrapping: cross-sectional SEM image showing the PDMS wrapping on the PP fiber yarns (a), conductivity (b) and Seebeck coefficient (c) of the pristine PP fibers and wrapped PP fibers in dry and wet conditions, the performance change of the PP fibers after multiple washing cycles (d).”

  1. How well is the ionic liquid additive bound in smart weaving and whether it does not escape from the fabric?

Response: In this work, the ionic liquid was used as additive in the wet-spinning dope. The weight ratio of this additive is typically below 2 wt%. Furthermore, we think most of the non-bonded ionic liquid was removed in coagulation bath. As a result, the ionic liquids make up very little of the final fiber, and if any remaining, they may form ionic bonds with PEDOT:PSS. Furthermore, after PDMS wrapping, the escape of ionic liquid should not be a problem.

  1. How the performance of the material is affected by multiple washing cycles?

Response: Thank you for your comment. The washing stability is very important when the smart electronic fabric is going to use in the real world. As we addressed to your 2nd comment, the stability of the fibers was improved by PDMS wrapping, and this is also beneficial for the washing stability as the wrapping protects the fibers in water. We implemented a test for wash stability, simulating the washing of fabrics by stirring the fibers in water for several hours. The results were shown in Figure 7d, and the discussion were provided in the main text:

  “We then washed the wrapped fibers several times in water. As shown in Figure 7d, after 5 cycles of washing, the conductivity and thermoelectric coefficient of the fiber remain above 95% of the original value. The margin loss may be due to the penetration of trace amounts of water molecules into the fiber from the crack or junction. Nevertheless, we believe that PDMS wrapping can significantly improve the working stability of PP fibers, providing a material basis for the realization of multifunctional smart fabrics.”

  1. Please rearrange the scheme of making the glove in the experimental part.

Response: We have rearranged the scheme of making glove in the experimental part as:

“2.4. Preparation of smart textiles and stability testing

Smart fabrics with thermal sensing and Joule heating functions were manufactured by weaving PEDOT:PSS microfiber onto commercial gloves. Firstly, ten micro-fibers are twisted to form a stable fiber yarn; Then, multiple fiber yarns were sewn onto the glove with a sewing needle. The sensing unit on the index finger consisted of 3 strands of yarn that detect temperature changes when grasping objects. The two ends of the yarn were placed on the inside and outside of the glove, fixed with silver paste and copper wire, and connected to the thermoelectric potential measurement system. The sensing unit on the back of the glove consisted of 6 strands of yarn to detect the ambient temperature and provide heating. One end of the bundle was positioned close to the skin, while the other end was exposed to the environment to sense the temperature difference between the environment and the skin of the hand.

To make the temperature sensor array, six bundles of yarn were sewn onto the fabric in the arrangement shown in Figure 6, while using silver paste and copper wire to connect both ends of each bundle to a multichannel voltage signal acquisition board.

In order to improve the working stability of the fibers, a thin layer of PDMS was wrapped on the surface of a yarn bundle consisting of 20 PP fibers. The twisted yarn bundle was slowly pulled through the freshly formulated PDMS precursor (Sylgard 184). After standing vertically for 15 minutes, the fiber bundles were treated in an oven at 80 °C for half an hour. The moisture stability of the yarn tested by dripping water in the middle of the yarn when the conductivity and Seebeck coefficient were measured. The PDMS-wrapped yarn was placed in water and stirred for one hour at a rate of 500 rpm for one wash cycle. After each wash, the conductivity of the fibers and the Seebeck coefficient were measured.”

  1. The first paragraph of results and discussion section describes the procedure for producing fibers and will be included in the experimental part of the work.

Response: thank you for your suggestion. We have shifted this paragraph to experimental part.

  1. One paragraph of discussion of the results is really not much for a scientific paper.

Response: the last part was changed to conclusions.

Round 2

Reviewer 2 Report

I recommend the article for publication in its current form.